# Establishment, Maintenance, and Performance of the Cooperative Osteosarcoma Study Group (COSS)

**DOI:** 10.3390/cancers15051520

**Published:** 2023-02-28

**Authors:** Stefan S. Bielack, Leo Kager, Thomas Kühne, Thorsten Langer, Peter Reichardt, Claudia Blattmann, Matthias Kevric, Vanessa Mettmann, Benjamin Sorg, Stefanie Hecker-Nolting

**Affiliations:** 1Cooperative Osteosarcoma Study Group, Pediatrics 5 (Oncology, Hematology, Immunology), Center for Pediatric, Adolescent and Women’s Medicine, Stuttgart Cancer Center, Klinikum Stuttgart—Olgahospital, 70174 Stuttgart, Germany; 2Klinik für Kinder- und Jugendmedizin, Pädiatrische Hämatologie und Onkologie, Universitätsklinikum Münster, 48149 Münster, Germany; 3St. Anna Kinderspital, University Hospital for Pediatric and Adolescent Medicine of the Medical University, and St. Anna Children’s Cancer Research Institute (CCRI), 1090 Vienna, Austria; 4Department of Pediatric Oncology/Hematology, Pediatric University Hospital Basel, CH-4031 Basel, Switzerland; 5Hospital for Pediatric and Adolesent Medicine, Pediatric Hematology and Oncology, Rheumatology and Immunology, Universitity Hospital Schleswig-Holstein, Campus Kiel, 24105 Kiel, Germany; 6Onkologie und Palliativmedizin, Helios Klinikum Berlin-Buch, 13125 Berlin, Germany

**Keywords:** osteosarcoma, collaborative trial, registry, child, adolescent, adult

## Abstract

**Simple Summary:**

The Cooperative Osteosarcoma Study Group (COSS), active since the late 1970s, is a multi-institutional consortium representing Germany, Austria, and Switzerland. Its goals are to cure as many osteosarcoma patients as possible with as few late effects as achievable. Over the decades, COSS has amassed and followed more than 5000 affected patients into trials and its registry. This has allowed the group to perform many meaningful analyses. These have focused on specific trials as well as on particular patient-, tumor-, or treatment-related variables. Intergroup cooperation has further expanded knowledge generation about this rare disease, its variants, and some closely related malignancies. The present paper presents an overview over more than four decades of fruitful collaboration.

**Abstract:**

Introduction: Osteosarcoma treatment has benefitted greatly from collaborative research. This paper describes the history and accomplishments of the Cooperative Osteosarcoma Study Group (COSS), mainly dedicated to clinical questions, as well as remaining challenges. Materials and Methods: Narrative review of over four decades of uninterrupted collaboration within the multi-national German–Austrian–Swiss COSS group. Results: Since its very first prospective osteosarcoma trial starting in 1977, COSS has continuously been able to provide high-level evidence on various tumor- and treatment-related questions. This includes both the cohort of patients enrolled into prospective trials as well as those patients excluded from them for various reasons, followed in a prospective registry. Well over one hundred disease-related publications attest to the group’s impact on the field. Despite these accomplishments, challenging problems remain. Discussion: Collaborative research within a multi-national study group resulted in better definitions of important aspects of the most common bone tumor, osteosarcoma, and its treatments. Important challenges continue to persist.

## 1. Origins and Early COSS History



*“If we operate they die, if we don’t operate they die. This meeting should be concluded with prayers.”*
(Sir Stanford Cade, 1955)


Osteosarcoma arises in approximately 2–3/Mio. individuals per year. Adolescent males are most frequently affected, but it may affect all ages and both genders. It was an almost universally fatal disease until a therapeutic revolution manifested itself some fifty years ago. Surgery, overwhelmingly often in the form of an amputation, had been performed for this disease for decades. It was mostly not curative: patients soon thereafter succumbed to metastases to the lungs. In the early 1970s, systemic therapies active against micro-metastases were finally discovered [1,2]. A breakthrough towards a cure was achieved when active agents were combined and employed in an adjuvant [3,4] and, very soon thereafter, a neoadjuvant setting [5]. The formerly untreatable disease had suddenly become curable.

The news was soon heard in Germany [6], Austria [7], and Switzerland. Inspired by the first optimistic reports in a previously unequivocally fatal malignancy, the Cooperative Osteosarcoma Study Group, COSS, was founded. As any single center would encounter far too few individual patients to come up with any meaningful findings, visionary clinicians and scientists joined efforts far beyond national borders. Together, they performed the first of many joint, multi-center, multi-national osteosarcoma trials [8,9]. Thus, the first multi-national group dedicated to this disease emerged. This laid the foundation for more than 45 years of collaboration against osteosarcoma and related malignancies.

## 2. Structure of the COSS Group

### 2.1. Multi-Disciplinarity

COSS has always been a decidedly multi-disciplinary group. This was due to the obvious fact that it took more than only one specialty to treat osteosarcoma. By themselves, no single discipline could conquer the disease. Together, they stood a realistic chance. The group therefore invited all specialties required for affected patients. In addition to both pediatric and medical oncologists, this includes radiologists and nuclear medicine specialists, responsible for primary and metastatic tumor imaging and staging; pathologists, offering reference histology in each and every case; and tumor surgeons, responsible for diagnostic biopsies and tumor removal with, if at all feasible, wide [10] margins. In special tumor locations—for instance, the head and neck—site-specific specialists assist orthopedic surgeons. Thoracic surgeons assess the possibilities of removing pulmonary metastases. Radiation oncologists, including proton and heavy-ion specialists, explore treatment options for unresectable primary and metastatic lesions. Other experts are in charge of questions surrounding, for instance, molecular tumor biology, late effects of therapy, quality of life, or statistics.

### 2.2. Multi-Centricity

COSS was designed as a disease-related, inclusive network. Not merely pure science, but also the best achievable care for as many patients as possible, has been and still is a major impetus. Over the decades, more than 200 individual institutions have registered between one and several hundred patients with the group. This allowed compensation for limited local expertise—expected in such a rare disease—by that of the group as a whole. This was accomplished by an ever more elaborate consultation system (see below). A downside of this is that some institutions merely seek expertise, without wishing to contribute to knowledge generation. COSS has thus fought a never-ending battle to propagate the virtues of collaborative knowledge generation.

### 2.3. Multi-Nationality

COSS, run under the auspices of the German Society for Pediatric Oncology and Hematology (GPOH), has traditionally been open for all institutions from Germany, Austria, and Switzerland. Some sites from the Czech Republic and Hungary have also contributed patients to the international EURAMOS trial [11].

### 2.4. Patient Recruitment

The group has always followed an inclusive registration strategy. Any patient with an osteosarcoma and some closely related tumors was eligible for registration into a disease-oriented registry. Although the clear majority of registered individuals stem from pediatric institutions, age has never been an exclusion factor. With the years and decades, the COSS registry has grown into the largest disease-related database worldwide (Figure 1).

Trials have at all times been limited to specific subpopulations who fulfilled relevant inclusion and exclusion criteria. Thus, important research questions were answered. However, prospective trials are bound to loose patient (sub-)groups. Information that could be learned from such would be lost. COSS has therefore implemented its patient registry with much less stringent entry criteria, run in parallel to any prospective trial from day one (Figure 2).

### 2.5. Long-Term Follow-Up and Cancer Survivorship

Risk-stratified long-term care, involving both tertiary care centers and the multi-disciplinary teams and general practitioners with whom they interact, is the group’s aim. Such was reported as the preferred model of care after cancer [12,13]. Cooperation is, however, challenged by the constant needs for communication, instruction, and training as well as continuous data sharing [13,14]. The recommendations for practical use presented here might serve as a tool to improve collaboration between multi-disciplinary teams and general practitioners. Such was highlighted in a recent Australian study, where highly prescriptive care plans from the oncologist/long-term follow-up clinic were the preferred mode of communication. However, such were often not provided [15].

## 3. Aims of the COSS Group

From the very beginning, the group has had two major aims: science and, no less important, providing every patient with the best available care.

Prospective clinical trials are the benchmark of clinical science. These have hence been a major focus. Additional information about osteosarcoma and related tumors was generated from the group’s clinical registry with far less stringent inclusion criteria.

Up-to-date clinical care to each registered patient is COSS’ second core aim. Clinical pathways were therefore implemented and a multi-disciplinary consultation service was put into place. Named specialists may be called upon to address individual questions that treating physicians may have. An interdisciplinary, real-life tumor board takes place once every week. The possibility to attend virtually is offered to the treating institutions.

## 4. Prospective COSS Trials

### 4.1. Groupwise Trials

The first of several published COSS trials, COSS-77, was a relatively small trial of adjuvant osteosarcoma chemotherapy. This trial proved both that chemotherapy worked in the disease and, notably, that multi-centric collaboration was possible in a rare cancer [8,9]. Neoadjuvant treatment was first introduced in the following trial, COSS-80. It could not demonstrate the superiority of one chemotherapy combination over another or a role for fibroblast interferon [9,16,17,18].

The follow-up trial, COSS-82, was a disappointment as far as outcomes were concerned, but a great success for the following generations of osteosarcoma patients. It was attempted to spare patients from treatment’s late effects. Particularly toxic substances were administered only post-operatively and only against tumors not responding to a less intensive regimen. As expected, the percentage of patients whose tumors responded to de-escalated therapy was lower than in the control arm, where individuals received intensive chemotherapy upfront. However, patients whose tumors did not respond to the less intensive regimen remained to have a very poor prognosis despite post-operative therapeutic intensifications. Consequently, all osteosarcoma patients now receive intensive therapy from day one [18,19].

Intra-arterial therapy was associated with great hope when it first made its entrance into osteosarcoma treatment. Cisplatin was proposed to be administered by this technique. COSS addressed its intra-arterial versus intravenous administration in a prospective, randomized trial, COSS-86. Early laboratory results by spectroscopy suggested caution: similar blood and intratumoral cisplatin concentrations were obtainable by either technique [20]. Clinical results later proved that neither the response rate nor the survival rate were improved with the intra-arterial technique [21,22,23,24]. Because of this and other trials [25,26], it was largely abandoned. The COSS-96 trial then attempted to introduce risk-based therapy but failed to be successful. A manuscript on its long-term results is in preparation.

### 4.2. Intergroup Trials

Acknowledging that even the largest individual efforts would need many years to answer randomized questions, several of the world’s leading osteosarcoma groups joined efforts in the largest prospective osteosarcoma study ever, EURAMOS-1 [27]. In addition to COSS, these were the Children’s Oncology Group (COG), the European Osteosarcoma Intergroup (EOI), and the Scandinavian Sarcoma Group (SSG). Together, they recruited well over 2200 patients [28]. Intergroup collaboration was not without its unique challenges. Each partner had to adapt. Nevertheless, all participants ultimately benefitted. In slightly over five years, two separate research questions were addressed. Firstly, postoperative therapeutic adaptations could not improve the poor prognosis of patients with a poor histological tumor response to upfront chemotherapy [29]. In the same trial, maintenance therapy with interferon-α was of no benefit for the remaining good responders [30]. Both conclusions remained valid with extended follow-up [11]. In addition to mere survival data, it was also possible to investigate the quality of life of the participating individuals [31,32]. Unfortunately, novel innovative research questions were lacking after EURAMOS-1 had closed, so that no follow-up trials materialized.

Osteosarcomas in older adults have always been largely uncharted territory. Prospective osteosarcoma trials have generally had an upper age limit of, at the most, 40 years. COSS, the Italian Sarcoma Group (ISG), and SSG addressed this deficit by jointly performing the only prospective trial ever in this cohort of patients. After addressing several challenging regulatory hurdles, the resulting European Over 40 Bone Sarcoma (EURO-B.O.S.S.) trial managed to include 218 adult osteosarcoma patients. The proposed treatment regimen differed from that in younger patients: primary surgery, while not advocated, was allowed. Drug doses were reduced and high-dose methotrexate was limited to patients with a non-response to doxorubicin, cisplatin, and ifosfamide. The study’s results proved that this prescribed therapy, while associated with substantial toxicity, was generally feasible. The resulting five-year overall survival rates of 66% for patients with seemingly localized disease and 22% for patients with primary metastases may be seen as new benchmarks for the age group [33].

## 5. COSS Registry

### 5.1. Rationale

Owing to more or less strict study entry and exclusion criteria, many patients would be lost to science if they were limited to prospective trials. The knowledge that could be gained from such patients and their unique disease situations is, however, substantial. This problem is addressed by the COSS registry with its very liberal inclusion and almost no exclusion criteria.

### 5.2. Recruitment

The COSS registry includes all patients from participating institutions, be they eligible for trials or not, with a diagnosis of osteosarcoma, whatever its grade might be. Some biologically related tumors, such as undifferentiated pleomorphic sarcomas or dedifferentiated or mesenchymal chondrosarcomas, may also be entered. Therapy according to published guidelines and guidance is suggested. However, the choice of specific therapeutic measures does not represent a prerequisite for recruitment.

### 5.3. Published Groupwise Analyses

By including all osteosarcoma patients and some others into its registry, COSS has been able to assess a wide variety of questions. Those based on the collectively gathered data are summarized here. A summary of all osteosarcoma results has been published [33]. For detailed analyses, this first and foremost included a report of all newly diagnosed osteosarcomas. This paper on prognostic factors of 1702 multi-modally treated patients with high-grade osteosarcomas set benchmarks. The tumor site and size, the presence or absence of primary disease spread, the tumor response to pre-operative chemotherapy, and, foremost, the surgical clearance of all diseased sites were proven as independent prognostic factors. According to pubmed.com (accessed on 22 February 2023), this has become the most referenced clinical osteosarcoma paper of all time [34].

Using the large COSS database, various specific tumor presentations and therapeutic details could be analyzed. For instance, tumor size was found to correlate closely with outcomes [35,36]. As for therapy, methotrexate (when given at a fixed dose of 12 g/m^2^) dose intensity was proven not prognostic in modern polychemotherapy protocols [37]. The dose intensity of received chemotherapy was the focus of another analysis. It was not found to correlate with treatment outcomes, neither were the received dose intensities of individual agents prognostic [38]. Great hopes were once associated with high-dose chemotherapy with blood stem cell rescue. A review of COSS patients treated in this way, usually for advanced disease situations, showed these to remain uncured even after this procedure, which was subsequently largely abandoned [39].

A very large cohort of 2847 COSS patients with high-grade central extremity osteosarcomas was recently screened for the presence of pathological fractures. They were present in 11.3% of patients. Pathological fractures correlated with the tumor site, histologic subtype, relative tumor size, and primary metastatic status. They were prognostic in adults but not in pediatric patients [40].

Primary metastases affect around 15% of osteosarcoma patients. They were another early research focus of the cooperative group [41]. A benchmark analysis of affected individuals clearly demonstrated that their number and the ability to achieve complete surgical remission were strong prognostic factors. In addition, the factors established as prognostic in localized disease held their value in the primary metastatic setting [42]. Skip metastases were not as negatively prognostic as previously assumed. This was true if the lesion affected the same bone as the primary tumor. Prognosis was inferior with trans-articular skip lesions [43].

The structure of data collection allowed the investigation of several of the more uncommon tumor sites, elucidating the roles of local and systemic therapies. Osteosarcomas of the hands [44] and feet [45] carried many of the same prognostic factors as did the more common long-bone primaries. Osteosarcomas affecting various locations in the axial skeleton shared many of those characteristics, with some distinct differences. Arising in somewhat older patients, they responded far less favorably to chemotherapy. Most importantly, however, surgical remission was found to be much more difficult to achieve. This resulted in a far inferior prognosis [46,47,48,49]. Again demonstrating the benefits of registering all osteosarcoma patients, COSS was also able to take a detailed look at craniofacial osteosarcomas. The role of chemotherapy at this site is often considered far less pivotal than with tumors elsewhere. COSS detected clues to its efficacy, without convincingly providing a definitive answer [50].

Osteosarcoma arising as a second or later malignancy was long thought to carry an almost uniformly fatal prognosis. A COSS analysis of 30 affected patients could, for the first time, prove this wrong. The predilection of secondary tumors for the axial skeleton, a consequence of former irradiation, remained challenging [51]. Even osteosarcoma arising after bone marrow transplantation, again often secondary to former radiotherapy, proved to be treatable [52,53]. The combination of osteosarcoma and a phyllodes tumor of the breast was observed most often in female patients affected by Li–Fraumeni syndrome [54]. Moreover, several cases of osteosarcoma in patients affected by Rothmund–Thomsen syndrome could be analyzed. In particular, affected individuals were evaluable for their chemotherapy tolerance [55].

The discussion about which type of therapy is to be employed, an osteosarcoma regimen or rather a soft-tissue sarcoma regimen, surrounds extraosseous osteosarcoma. A review of the COSS experience demonstrated favorable results with osteosarcoma-based regimens. It must, however, be noted that the COSS patients analyzed were far younger than the average extraosseous osteosarcoma patient [56].

Osteosarcoma most often affects children or adolescents. The typical preponderance of males was not evident in the youngest patients below the age of five years at diagnosis. Otherwise, they generally seemed to behave as expected [57].

Misdiagnosis and then mistreatment of osteosarcoma as some other, often benign tumor is one of the most dreaded mishaps of oncologists. Hesitancy often precludes the reporting of such diagnostic failures. COSS recently reported on such patients and could prove that uncommon sites of tumor presentation were at a particular risk for misinterpretation. Systemic spread occurring during the lag time between incorrect and correct diagnoses was observed. Some affected patients were still cured once appropriate therapy was finally initiated [58].

COSS’ unlimited follow-up allowed a detailed look at those unfortunate patients who developed disease recurrences. As assumed, these were most often pulmonary, followed by the distant bones and local failures. The timing and number of metastases correlated with outcomes. Achieving a (second) complete surgical remission was essentially found to be a prerequisite for a cure. The use of second-line chemotherapy seemed prognostically favorable, but its influence was limited [59]. Even second and subsequent recurrences could be investigated with large patient numbers. The ability to achieve renewed surgical remissions once again emerged as pivotal. Some patients became long-term survivors despite multiple recurrences. Renewed chemotherapy may again have contributed, but within very narrow limitations [60]. Regarding special metastatic sites, an analysis of distant osseous recurrences demonstrated these to be treatable if solitary [61]. Focusing on the local site as the region of recurrence, 76 out of 1355 analyzed patients with extremity or axial osteosarcomas developed this complication. Not participating in a clinical trial, pelvic primaries, limb-sparing surgery, soft-tissue infiltration beyond the periosteum, a poor response to neo-adjuvant chemotherapy, a failure to complete the planned chemotherapy protocol, and a biopsy at a center other than the one performing the definitive procedure were significant predictors of an increased local recurrence risk. No differences were obvious for varying surgical margin widths. Surgical treatment at centers with a small patient volume and more than one surgical procedure of the primary tumor area were significantly associated with a higher rate of ablative surgery [62].

As for long-term outcomes after a rotationplasty, these were the results of a recent patient survey. It proved this procedure to be a realistic therapeutic option for eligible patients, with few revision procedures needed even long-term [63].

In addition to classical osteosarcoma, the group has always followed some other tumors considered biologically related. Undifferentiated pleomorphic sarcoma (UPS, formerly malignant fibrous histiocytoma, MFH) of bone, the focus of a European joint analysis, responded to many of the same treatment principles as osteosarcoma itself. While the histologic response rate to chemotherapy was worse, the overall outcome was nevertheless quite similar [64]. The COSS registry’s inclusion of tumors biologically similar to osteosarcoma allowed the group to contribute to the dedifferentiated chondrosarcoma cohort of the EURO-B.O.S.S. study. It could thus be demonstrated how patients (aged 40–65 years) suffering from such a malignancy fared when treated according to an up-to-date, multi-disciplinary approach, setting a new benchmark for the disease [65].

The group’s efforts never ended with successful antineoplastic treatment. As early as 1983, its toxicities also came into focus [66]. A major step forward was the foundation of the Late Effects Surveillance System (LESS), GPOH’s collaborative late effects study group, to which COSS contributes relevant data regularly [67]. Analyses including former osteosarcoma patients have focused on doxorubicin’s cardiotoxicity [68], the platinum analogues ototoxicitis [69,70], cisplatinum, and ifosfamide’s nephrotoxicity [71,72], the thyroid’s function after therapy [73], and the long-term immunity of these heavily treated patients [74,75]. Lately, efforts to better define a treatment’s late effects have been extended into a European collaboration within the PANCARE consortium. As a first step, it was thus possible to amass sufficient patients for analyses of cisplatin’s ototoxicity [76,77].

## 6. Adolescent and Young Adult Oncology

Osteosarcomas are among the most frequent malignancies affecting adolescents and young adults (AYA). Anyone with even a remote interest in osteosarcomas will therefore necessarily need to address the specific challenges associated with this transitional period between childhood and adult life. Historically, AYA have been lost between pediatric and medical oncology, with very limited interaction between these. Often, this resulted in completely different approaches to the same diseases. Consequently, individuals from the COSS group have had roles in the foundation and leadership of both national and European AYA working groups, bridging the relevant national [78] and continental [79] pediatric and medical oncology societies. These collaborations have led to several pivotal publications in the field of osteosarcoma in AYAs [80] and AYA oncology in general [81,82].

## 7. Intergroup Collaboration Using Anonymized Data

Anonymized patient baseline, treatment, and outcome data from COSS were entered into a variety of retrospective collaborative intergroup analyses, allowing for adequate patient numbers. It was thus possible to gather a very large cohort of patients for potential prognostic differences between children, adolescents, and young adults with osteosarcoma and to demonstrate that the youngest patients had a more favorable prognosis [83]. This form of collaboration also helped to describe the successes and knowledge gaps in older adults with osteosarcoma [84]. Periosteal osteosarcomas were shown to require meticulous local therapy. A role for systemic chemotherapy could not be documented [85]. A detailed analysis of 266 extraosseous osteosarcomas defined both similarities and prognostic and therapeutic distinctions from their osseous counterparts [86]. Those patients who developed very late osteosarcoma recurrence were the focus of one [87], and the toxicity of high-dose methotrexate, a pivotal drug in the disease, of another large collaborative analysis [88].

European collaboration also helped to better define the principles of local and systemic therapy in mesenchymal chondrosarcoma [89]. The prevalence and outcomes of secondary malignancies after a diagnosis of a sarcoma of any type and their relation to predisposing factors were the topics of a collaboration between GPOH’s bone and soft tissue sarcoma groups [90]. Rare bone tumors other than osteosarcoma could be analyzed in part thanks to COSS’s contribution to Germany’s network of rare pediatric tumors [91,92].

## 8. Bringing European Researchers Together

Osteosarcoma treatment has seen little prognostic improvement over the past few decades. Trying to end this stalemate, COSS has had its role in bringing Europe’s clinical and laboratory osteosarcoma researchers together to discuss their current projects and future plans [93,94,95]. Individuals charged with leading roles within the COSS group have had rules in drafting both national [96] and European [97] osteosarcoma guidelines. The group was also influential in drafting essential requirements for the quality care of affected patients on a European level [98].

## 9. Past, Current, and Future Challenges to Collaboration

### 9.1. Maintenance of Multi-Disciplinary Collaboration

Relevant publications with COSS contribution are summarized in Table 1. The successful running of such a large multi-disciplinary, multi-institutional study group is no one-time effort, but poses its own, perpetual challenges. Potential study group members must understand the benefits of being part of the endeavor and accept the obligations coming with their participation. For them, the benefits must clearly outweigh the costs. It remains a constant struggle to convince physicians that COSS and especially its consultation service are no easy way to substitute for inadequacies, but that this requires input on their part. However, this seems by now to be widely understood throughout the participating countries, with notable exceptions.

### 9.2. Ever Increasing Regulatory Demands

The early years of COSS cooperation had lax ethical requirements and fledgling regulatory demands. This has definitely changed. Any patient-related project is now the subject of very strict ethical guidelines, designed to protect patients from harm through unjustified or ill-considered interventions. As Germany is part of the European Union and the COSS study center is located in Germany, one of its member states, it also bears the full brunt of the EU Good Clinical Practice directive 2001/20. Mainly designed to protect patients, it also brought with it an unparalleled bureaucratic workload and serious financial consequences. The directive’s negative consequences on investigator-initiated research are unquestionable: designed with the well-meaning intention to protect participants in clinical trials, it may rather prevent trials from ever opening. Recent legislative changes designed to ease this situation promise long-awaited relief, but this only time will tell.

### 9.3. Financial Sustainability

The financial sustainability of trial groups such as COSS is a constant challenge. Its trials and registries have been funded by Deutsche Krebshilfe (DKH), Deutsche Forschungsgesellschaft (DFG), the European Science Foundation (ESF), charities, and others. A major step forward was Germany’s public insurance companies recently agreeing to reimburse GPOH’s study centers for their remote patient consultations. It remains open if the recent ERN PaedCan initiative to reimburse for consultations from other EU member states will be met with success. Its online reference system may provide quick answers, but also places considerable obligations on those asking for advice.

## 10. Current and Future COSS Projects

COSS has had its part in recent international efforts to define the current treatment principles, and the solved and the open questions in osteosarcoma [100,101]. However, the group has had no open trial for its main patient cohort, young patients with osteosarcoma, for over 10 years. A paucity of new, efficacious agents and ever-increasing regulatory demands have led to a certain standstill. The COSS registry, however, is ongoing. An updated version recently received ethical approval. It now includes biological questions and a centralized tumor bank.

As one possible way forward, COSS has entered into several large-scale, international collaborations. One of these, FOSTER (Fight Osteosarcoma Through European Research), is a Pan-European effort to gather experts from all European countries. Organized in work packages, FOSTER seeks to answer relevant tumor-biological and clinical questions. Ultimately, it may even develop multi-institutional, international trials. On a global scale, multiple groups interact in the Harmonization International Bone Sarcoma Consortium (HIBISCus), led by the University of Chicago. This seeks to collect existing data into a harmonized database for later analysis [102]. COSS’ ultimate goal, however, is to once again perform prospective, randomized osteosarcoma trials.

## 11. Conclusions

The Cooperative Osteosarcoma Study Group, COSS, has been witness to unprecedented progress in the fight against a rare cancer, as well as to highly frustrating prognostic stagnation. It has been able to contribute pivotal data to the field and novel answers to specific questions. Further advances will now probably or even definitely require completely novel approaches, without ever forgetting what made osteosarcoma therapy successful in the first place. The group will address these challenges under new leadership and with renewed vigor. Its ultimate goal remains unaltered: one day, no patient will have to die from osteosarcoma.

## Figures and Tables

**Figure 1 cancers-15-01520-f001:**
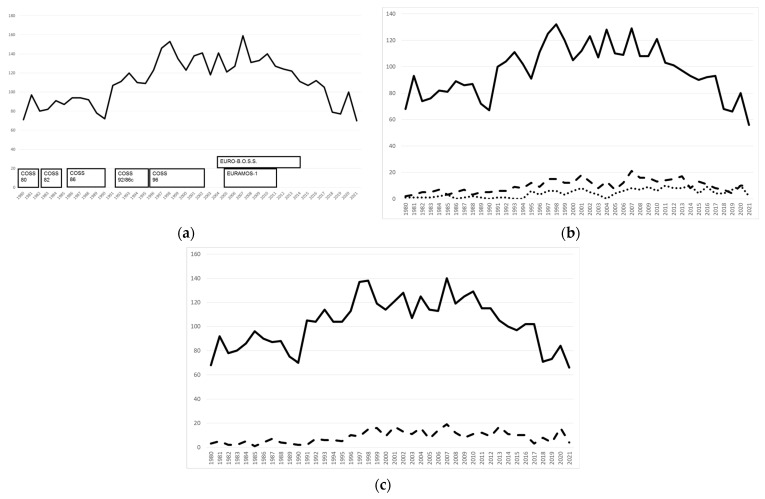
Recruitment per year of 4657 previously untreated Cooperative Osteosarcoma Study Group osteosarcomas 1980–2001. Patients registered > 1 year after biopsy excluded. (**a**)—all 4657 patients. (**b**)—by primary tumor site: limb (*n* = 4071, solid line) vs. trunk (*n* = 404, dashed line) vs. head and neck (*n* = 177, dotted line); 5 primary sites unknown. (**c**)—by grade of malignancy: high-grade central (*n* = 4136, solid line) vs. others (*n* = 337, dashed line); head and neck osteosarcomas excluded; 8 grades not documented.

**Figure 2 cancers-15-01520-f002:**
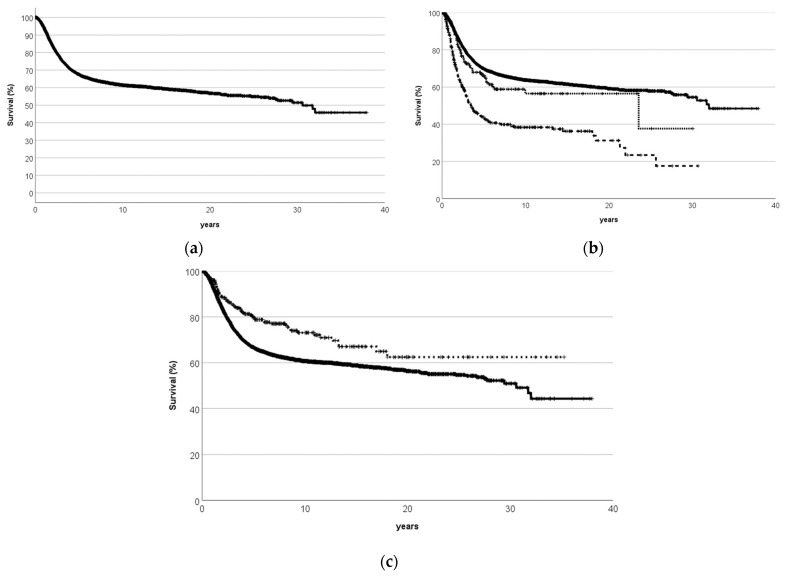
Survival probability of previously untreated Cooperative Osteosarcoma Study Group patients with osteosarcomas 1980–2001. Patients registered >1 year after biopsy excluded. Median follow-up 5.04 (0.003—37.96) years from diagnostic biopsy. (**a**)—all 4657 patients. (**b**)—by primary tumor site: limb (*n* = 4071, solid line) vs. trunk (*n*= 404, dashed line) vs. head and neck (*n* = 177, dotted line); 5 primary sites unknown. Osteosarcomas of the trunk had an inferior survival probability to either those of the extremities or the head and neck (*p* < 0.001, log-rank test, respectively). (**c**)—by grade of malignancy: high-grade central (*n* = 4136, solid line) vs. others (*n* = 337, dashed line); head and neck osteosarcomas excluded; 8 grades not documented (*p* < 0.001, log-rank test).

**Table 1 cancers-15-01520-t001:** Publications on oncological topics by the COSS group or with COSS contribution. Results relate to osteosarcoma and to COSS patients as long as not explicitly stated otherwise.

Publication	Ref.	Doi	Topic
Groupwise trials			
Winkler1982	[8]	10.1055/s-0028-1105579	COSS-77: first adjuvant trial results
Winkler 1983	[12]	10.1007/BF00625042	COSS-80: preliminary trial results
Winkler 1984	[13]	10.1200/JCO.1984.2.6.617	COSS-80: final trial results
Purfürst 1985	[9]	10.1055/s-2008-1033974	COSS-77 and -80: updated trial results
Winkler 1988	[15]	10.1200/JCO.1988.6.2.329	COSS-82: final trial results
Bielack 1989	[14]	10.1055/s-2008-1026715	COSS-80 and COSS-82: updated trial results
Winkler 1990	[17]	-	COSS-86: preliminary trial results
Winkler 1990	[18]	10.1002/1097-0142(19901015)66:8<1703::aid-cncr2820660809>3.0.co;2-v	COSS-86: final trial results
Bieling 1996	[19]	10.1055/s-2007-1025433	COSS-86: preliminary trial results
Fuchs 1998	[20]	10.1023/a:1008391103132	COSS-86: updated trial results
Bielack 2009	[30]	10.1007/978-1-4419-0284-9_15	COSS: pooled results
Intergroup trials			
Marina 2009	[23]	10.1007/978-1-4419-0284-9_18	EURAMOS-1: design
Whelan 2015	[24]	10.1093/annonc/mdu526	EURAMOS-1: pre-randomization results
Bielack 2015	[26]	10.1200/JCO.2014.60.0734	EURAMOS-1: poor responder results
Marina 2016	[25]	10.1016/S1470-2045(16)30214-5	EURAMOS-1: good responder results
Ferrari 2018	[29]	10.5301/tj.5000696	EURO-B.O.S.S S.: osteosarcoma results (>40 years)
Smeland 2019	[11]	10.1016/j.ejca.2018.11.027	EURAMOS-1: updated trial results
Calaminus 2019	[27]	10.1016/j.ejca.2022.03.018	EURAMOS-1: quality of life methodology
Hompland 2021	[61]	10.1016/j.ejca.2021.04.017	EURO-B.O.S.S: dedifferentiated chondrosarcomas in patients 41–65
Budde 2022	[28]	10.1016/j.ejca.2022.03.018	EURAMOS-1: quality of life results
Patient-related variables and outcomes			
Grimer 2003	[80]	10.1016/s0959-8049(02)00478-1	EMSOS: osteosarcoma over the age of forty
Bielack 2003	[48]	10.1038/sj.bmt.1703864	Osteosarcoma after bone marrow transplantation
Kager 2010	[53]	10.1002/cncr.25287	Osteosarcoma in very young children
Collins 2013	[79]	10.1200/JCO.2012.43.8598	Intergroup meta-analysis: younger vs. older patients with osteosarcoma
Bielack 2015	[50]	10.1097/MPH.0000000000000197	Osteosarcoma and phyllodes tumor
Zils 2015	[49]	10.1097/MPH.0b013e3182a2719c	Osteosarcoma after bone marrow transplantation
Zils 2015	[51]	10.3109/08880018.2014.987939	Osteosarcoma in Rothmund–Thomson syndrome
Gotta 2022	[59]	10.1055/a-1681-1916	Questionnaire: long-term function and quality of life with a rotationplasty
Tumor-related variables and outcomes			
Bieling 1996	[33]	10.1200/JCO.1996.14.3.848	Initial tumor size and prognosis
Rehan 1993	[32]	10.1055/s-2007-1025228	Initial tumor size and prognosis
Bielack 1995	[42]	10.1002/mpo.2950240103	Osteosarcoma of the trunk
Bielack 1999	[47]	10.1200/JCO.1999.17.4.1164	Osteosarcoma as secondary malignancy
Bielack 2002	[31]	10.1200/JCO.2002.20.3.776	Prognostic factors in osteosarcoma
Ozaki 2002	[43]	-	Osteosarcoma of the spine
Ozaki 2003	[44]	10.1200/JCO.2003.01.142	Osteosarcoma of the pelvis
Daecke 2005	[41]	10.1245/ASO.2005.06.002	Osteosarcoma of the hand and forearm
Jasnau 2008	[46]	10.1016/j.oraloncology.2007.03.001	Craniofacial osteosarcoma
Zils 2013	[45]	10.1093/annonc/mdt154	Osteosarcoma of the mobile spine
Schuster 2018	[40]	10.1155/2018/1632978	High-grade osteosarcomas of the foot
Kelley 2020	[36]	10.1200/JCO.19.00827	Pathological fracture and prognosis
Hecker-Nolting 2022	[54]	10.1007/s00432-022-04156-1	Osteosarcoma pre-diagnosed as another tumor
Primary and secondary metastatic disease			
Winkler 1989	[37]	10.1159/000216608	Primary metastatic osteosarcoma
Kager 2003	[38]	10.1200/JCO.2003.08.132	Primary metastatic osteosarcoma
Kempf-Bielack 2005	[55]	10.1200/JCO.2005.04.063	Osteosarcoma relapse after combined modality therapy
Kager 2006	[39]	10.1200/JCO.2005.04.2978	Primary skip metastases
Hauben 2006	[83]	10.1016/j.ejca.2005.09.032	Intergroup analysis: late osteosarcoma relapses
Bielack 2009	[56]	10.1200/JCO.2008.16.2305	Second and subsequent osteosarcoma recurrences
Franke 2011	[57]	10.1002/pbc.22864	Solitary skeletal osteosarcoma recurrences
Andreou 2011	[58]	10.1093/annonc/mdq589	Local osteosarcoma recurrences
Osteosarcoma variants and non-osteosarcomas			
Bielack 1999	[60]	10.3109/17453679908997824	EMSOS: undifferentiated pleomorphic sarcoma
Grimer 2005	[81]	10.1016/j.ejca.2005.04.052	EMSOS: periosteal osteosarcoma
Goldstein-Jackson 2005	[52]	10.1007/s00432-005-0687-7	Eextraskeletal osteosarcoma
Brecht 2014	[88]	10.1002/pbc.24997	STEP: rare malignant pediatric tumors
Frezza 2015	[85]	10.1016/j.ejca.2014.11.007	EMSOS: mesenchymal chondrosarcoma
Longhi 2017	[82]	10.1016/j.ejca.2016.12.016	EMSOS: extraskeletal osteosarcoma
Anti-tumor drugs			
Bielack 1989	[16]	10.1007/BF00257446	Tumor tissue cisplatin levels after i.a. vs. i.v. infusion
Graf 1994	[99]	10.1200/JCO.1994.12.7.1443	Methotrexate pharmacokinetics and prognosis
Sauerbrey 2003	[35]	10.1038/sj.bmt.1703023	High-dose chemotherapy in relapsed osteosarcoma
Widemann 2004	[84]	10.1002/cncr.20255	Meta-analysis: high-dose methotrexate-induced nephrotoxicity
Eselgrim 2006	[34]	10.1002/pbc.20608	Dose intensity of chemotherapy and outcomes
Side effects of therapy			
Jürgens 1983	[62]	10.1007/BF00625045	Toxicity of osteosarcoma chemotherapy
Langer 2004	[63]	10.1002/pbc.10325	LESS: overview of late toxicity in sarcoma patients
Stöhr 2005	[65]	10.1081/cnv-200055951	LESS: cisplatin-induced ototoxicity in osteosarcoma
Paulides 2006	[64]	10.1002/pbc.20492	LESS: doxorubicin-induced cardiomyopathy in sarcoma
Stöhr 2007	[67]	10.1002/pbc.20812	LESS: nephrotoxicity of cisplatin and carboplatin in sarcoma
Stöhr 2007	[68]	10.1002/pbc.208	LESS: ifosfamide nephrotoxicity in sarcoma
Paulides 2007	[69]	10.1111/j.1365-2265.2007.02813.x	LESS: thyroid function in pediatric and young-adult sarcoma
Paulides 2010	[70]	10.1055/s-0030-1249609	LESS: Immunity against tetanus and diphtheria after childhood sarcoma
Paulides 2011	[71]	10.1016/j.vaccine.2010.12.084	LESS: antibodies against tetanus and diphtheria after childhood sarcoma
Nitz 2013	[66]	10.3892/ol.2012.997	LESS: cisplatin- and carboplatin-mediated ototoxicity in sarcoma
Langer 2020	[72]	10.1016/j.dib.2020.106227	PanCareLIFE: association of pharmacogenetic markers and platinum ototoxicity
Langer 2020	[73]	10.1016/j.ejca.2020.07.019	PanCareLIFE: genetic markers and platinum-induced ototoxicity
Kube 2022	[86]	10.1002/cncr.34110	COSS, CESS, and CWS: secondary malignancies after sarcomas
Guidelines/guidance/consensus papers			
Wilhelm 2014	[76]	10.1093/annonc/mdu153	ENCCA WP17-WP7: consensus on teenagers/young adults with bone sarcoma
Andritsch 2017	[94]	10.1016/j.critrevonc.2016.12.002	ECCO: essential requirements for quality sarcoma care
AWMF 2021	[92]	-	Expert consensus: German osteosarcoma guidelines
Strauss 2021	[93]	10.1016/j.annonc.2021.08.1995	ESMO-EURACAN-GENTURIS-ERN PaedCan: European sarcoma guidelines
International reviews			
Isakoff 2015	[95]	10.1200/JCO.2014.59.4895	Osteosarcoma treatment and a collaborative pathway to success
Beird 2022	[96]	10.1038/s41572-022-00409-y	Osteosarcoma

Abbreviations: ref. = reference, doi = digital object identifier, COSS = Cooperative Osteosarcoma Study Group, EURAMOS = European and American Osteosarcoma Study Group, EURO-B.O.S.S S. = EUROpean Bone Over Forty Osteosarcoma Study, EMSOS = European Musculo-Skeletal Oncology Society, STEP = Register Seltene Tumor-Erkrankungen in der Pädiatrie, LESS = Late Effects Surveillance System, PanCareLIFE = a pan-European consortium that addresses survivorship issues, CESS = Cooperative Ewing Sarcoma Study Group, CWS = Cooperative Weichteilsarkom-Studiengruppe, ENCCA WP17-WP7 = European Network for Cancer in Childhood and Adolescence Work Package 17—Work Package 7, ECCO = European Cancer Organisation, AWMF = Arbeitsgemeinschaft der Wissenschaftlich-Medizinischen Fachgesellschaften, ESMO-EURACAN-GENTURIS-ERN PaedCan = European Society for Medical Oncology—European Reference Network on Rare Adult Solid Cancers—European Reference Network on Genetic Tumor Risk Syndromes—European Reference Network on Paediatric Cancers.

## Data Availability

The data presented in this study are available on request from the corresponding author. The data are not publicly available due to lack of ethical approval.

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
