# Peer review of "Establishment, Maintenance, and Performance of the Cooperative Osteosarcoma Study Group (COSS)"

_cancers, 2023, doi:10.3390/cancers15051520_

Round 1
Reviewer 1 Report
The authors present an interesting manuscript that provides very adequate information. However, the authors need to improve important aspects:
-First of all, the title of the manuscript must be more specific.
-The references used must be updated.
-Formal aspects must be improved and adapted to the standards and forms of this Journal. Please check the references.
-Section 1 should be improved. The authors should make a more appropriate introduction to the spirit of this Journal.
-The figures should improve the quality.
-Tables should be better described and adapted to the format of this Journal.
-The authors should discuss the novelty and the future path.
-Authors should very specifically improve the use of English grammar with experts.
Author Response
The authors present an interesting manuscript that provides very adequate information. However, the authors need to improve important aspects:
-First of all, the title of the manuscript must be more specific.
The title was rephrased and now reads “Establishment, maintenance, and performance of the Cooperative Osteosarcoma Study Group (COSS)” (see comment to reviewer 3’s appropriate comment.
-The references used must be updated.
I am not quite sure I understand this comment, as the references are quite up-to-date. What might the reviewer like to see included?
-Formal aspects must be improved and adapted to the standards and forms of this Journal. Please check the references.
The references were checked and the only possible deviation from any journal’s usual “formal aspects” that could be detected was the naming all authors. The journal’s instructions, available in the internet, do not specify how many authors should be named - or I missed the instruction if there was one. However, the text was altered: Now, only the first three authors of each paper are mentioned in the literature section, followed by “et al.”. Hopefully, this is what the reviewer meant!
-Section 1 should be improved. The authors should make a more appropriate introduction to the spirit of this Journal.
I am not sure I grasped the true “spirit of the Journal”. Please advise if any alterations are needed. As is, the introduction – to my belief – gives a suggestion of what is to come.
-The figures should improve the quality.
Sorry, but we hope the figures do indeed improve the quality of the manuscript. They provide important information not detailed in the text. Please be more specific about any modifications or additions you might like to see!
-Tables should be better described and adapted to the format of this Journal.
The table is described succinctly and I do not quite grasp what the reviewer might expect. Please advise.
(Following a comment by reviewer 2, table 1 was also resorted, as were the explanations in the legend).
-The authors should discuss the novelty and the future path.
This report covers the history and achievements of COSS until the present time, under the first two COSS chairpersons. The second of these, the first author of this manuscript, resigned with the start of the new year, 2023. A replacement is due for election by the next GPOH General Assembly. The new chair will make her/his own plans for the future. The authors are neither aware of those nor do they want to interfere. This is a manuscript about accomplishments so far.
-Authors should very specifically improve the use of English grammar with experts.
The text was carefully reviewed and revised with the assistance of a U. S. highschool graduate. We hope this will suffice!
Reviewer 2 Report
Title: “Creation, maintenance, performance, and future of a large internationalosteosarcoma study-group. A report by the Cooperative Osteosarcoma Study
Group (COSS)” Authors: Stefan S. Bielack, Leo Kager, Thomas Kühne, Thorsten Langer,
Peter Reichardt, Claudia Blattmann, Matthias Kevric, Vanessa Mettmann,
Benjamin Sorg and Stefanie Hecker-Nolting
Summary:
This manuscript contains a report of the accomplishments of an internationalworking group (COSS) that has worked extensively on cases of osteosarcoma
patients. The working methods and advances of the COSS are clearly
summarized and provide a good overview, including for clinicians.
Several points are listed below:
1) The type of publication planned (Report?) should be added above the title. 2) The title reads a bit awkward. The authors might consider simplifying it, forexample, "A report on the establishment, maintenance, performance, and
future of the major international Cooperative Osteosarcoma Study Group
(COSS)".
3) The introduction should mention how many affected osteosarcoma patientsthere are worldwide and the proportions between children, adolescents and
adults. 4) Figure 1a: Please add meaning to the lines, analogous to the other legends. 5) Chapters 6, 7 and 8 seem a bit out of place. A different structure could be
considered here. 6) Table 1 data is shifted in the current presentation and should be ordered. 7) A listing of "Author Contributions" should be included at the end of the
report. 8) The bibliography consists of over 70% self-citations. Understandable since
it summarizes the history of COSS, yet a one page report. It would be
desirable to see more comparison of the work of COSS with the results of
other working groups.
Author Response
This manuscript contains a report of the accomplishments of an international working group (COSS) that has worked extensively on cases of osteosarcoma patients. The working methods and advances of the COSS are clearly summarized and provide a good overview, including for clinicians.
Several points are listed below:
1) The type of publication planned (Report?) should be added above the title.
Done as requested.
2) The title reads a bit awkward. The authors might consider simplifying it, for example, "A report on the establishment, maintenance, performance, and future of the major international Cooperative Osteosarcoma Study Group (COSS)".
Taking the reviewer’s suggestion into account, the title was altered. It now reads: “Establishment, maintenance, and performance of the Cooperative Osteosarcoma Study Group (COSS)”
3) The introduction should mention how many affected osteosarcoma patients there are worldwide and the proportions between children, adolescents and adults.
Sentences on the incidence and an appropriate reference were now included. The revised section reads: “Osteosarcoma arises in approximately 2-3/Mio. individuals per year. Adolescent males are most frequently affected but it may affect all ages and both genders. It was an almost universally fatal disease ...... ”
4) Figure 1a: Please add meaning to the lines, analogous to the other legends.
Sorry! An incorrect figure was inadvertently provided, thereby leaving the legend incomplete. The correct figure is now included and the legend should “fit”. (I did not know how to “color” the old figure in red and the new figure in green, so the exchange is not evident from the colored version of the revised manuscript).
5) Chapters 6, 7 and 8 seem a bit out of place. A different structure could be considered here.
We assume the reviewer refers to the chapters “Adolescent and young adult oncology”, “Intergroup collaboration using anonymized data”, and “Bringing European researchers together”. We agree that each of those differs in content from other chapters in its own particular way. However, each reflects important aspects of COSS’ efforts to improve the fate of osteosarcoma-patients in the German-speaking countries, in Europe, and worldwide. After very careful deliberation, we have therefore decided to leave them in this manuscript. Those readers who might find the aspects they cover less interesting than the others will skip to the next chapter without larger effort.
6) Table 1 data is shifted in the current presentation and should be ordered.
In response to this request and a somewhat similar request by reviewer 3, the table was completely re-organized. Please see the revised manuscript for details, thank you!
7) A listing of "Author Contributions" should be included at the end of the report.
Done. This was added at the end of the manuscript.
8) The bibliography consists of over 70% self-citations. Understandable since it summarizes the history of COSS, yet a one page report. It would be desirable to see more comparison of the work of COSS with the results of other working groups.
Sorry, but complying with this request would completely alter the scope of this manuscript. This is not about whether COSS is better or worse than other groups: The manuscript solely intends to summarize the group’s accomplishments. Complying with this request would mean a completely different paper. We have therefore decided to rather not. We hope this will be acceptable.
Reviewer 3 Report
fig 1 unclear. please provide a more detailed legend.
I would suggest a further analysis (graphic also) regarding different period of patients enrollment (eg subsequent decades)
Table 1. I suggest to report outcomes (eg survival, rate of amputation) rather than mere bibliography
Also, I would split this table into COSS results and COSS collaborations.
Please provide a comparison with similar study-groups on OS.
I would further discuss on future directions for the group.
Author Response
fig 1 unclear. please provide a more detailed legend.
This comment probably relates to figure 1a, this was altered (see above, comment to reviewer 2, comment 4). We once again apologize for our error!
I would suggest a further analysis (graphic also) regarding different period of patients enrollment (eg subsequent decades)
This information (changes of recruitment by time) can be seen in figures 1a, 1b, and 1c. Which information would the reviewer like to see added?
Table 1. I suggest to report outcomes (eg survival, rate of amputation) rather than mere bibliography
This would imply a completely different type of manuscript. Differences between recruitment strategies, resulting patient cohorts, non-systemic treatment details and the like would have to be explained to allow meaningful conclusions. This would be far beyond the scope of our current manuscript. It would also lead to excess text. This would exceed any allowable word counts by far. We therefore decided to leave the structure of the table unaltered. The interested reader might refer to the literature or to the authors for details.
Also, I would split this table into COSS results and COSS collaborations.
Please see the answer to reviewer 2’s somewhat similar request no. 6, above.
Please provide a comparison with similar study-groups on OS.
This manuscript is not about comparisons, but about COSS. A comparison to other groups would require a completely different paper.
I would further discuss on future directions for the group.
As mentioned above, in response to reviewer 1: This report covers the history and achievements of COSS until the present time, under the first two COSS chairpersons. The second of these, the first author of the present manuscript, resigned with the start of 2023. A replacement is due for election by the next GPOH General Assembly. The new chair will make her/his own plans for the future. The authors are neither aware of those nor do they want to interfere. This is a manuscript about what COSS accomplished so far.
Round 2
Reviewer 3 Report
Thank you to the Authors for their attempts to ameliorate this paper.
Despite their efforts, they did not answer to any of the previous comments satisfactorily. In particular, I still believe that a comparison of their network characteristics to other networks would be extremely important.
Actually, the paper is a mere report on what the Authors did.